# Activity in Group-Housed Home Cages of Mice as a Novel Preclinical Biomarker in Oncology Studies

**DOI:** 10.3390/cancers15194798

**Published:** 2023-09-29

**Authors:** Stéphane Terry, Céline Gommet, Anne-Cécile Kerangueven, Mickaël Leguet, Vincent Thévenin, Mickaël Berthelot, Laurent Begoud, Fanny Windenberger, Pierre Lainee

**Affiliations:** 1Research Department, Inovarion, 75005 Paris, France; stephane.terry@inovarion.com; 2Translational In Vivo Models—In Vivo Research Center Vitry, Sanofi Research and Development, 94403 Vitry-sur-Seine, France; celine.gommet@sanofi.com (C.G.); michael.leguet@sanofi.com (M.L.); vincent.thevenin@sanofi.com (V.T.); michael.berthelot@sanofi.com (M.B.); laurent.begoud@sanofi.com (L.B.); 3Biostatistics & Programming, Sanofi Research and Development, 94403 Vitry-sur-Seine, France; anne-cecile.kerangueven-ext@sanofi.com (A.-C.K.); fanny.windenberger@sanofi.com (F.W.)

**Keywords:** digital biomarkers, DVC^®^, home cage, translation, preclinical, animal model, oncology, chemotherapeutic agents, SCID model, drug development, welfare, toxicity and safety, behavior, cognitive impairment

## Abstract

**Simple Summary:**

Improving experimental conditions in preclinical animal studies is essential. Automated digital ventilated cages offer the advantage of continuous monitoring of animal locomotor activity in their home cage over a long period, improving animal welfare through the reduction in animal handling. The potential utility of this technology remains understudied and deserves investigation in the field of oncology. Here, we determined the utility of continuous assessment of locomotor activity using the DVC^®^ technology on the SCID (severe combined immunodeficiency) mouse model treated with reference oncology compounds. Our results showed that the use of the DVC^®^ locomotion index is able to identify significant deleterious effects on animal activity, while suggesting perturbations of animal behaviors that would not have been detected by staff clinical assessment over a limited daytime observation. The use of this technology also helped to detect the anticipated effects of the drugs tested, highlighting the potential of this digital biomarker to differentiate treatment adverse effects in preclinical oncology studies.

**Abstract:**

Background: Improving experimental conditions in preclinical animal research is a major challenge, both scientifically and ethically. Automated digital ventilated cages (DVC^®^) offer the advantage of continuous monitoring of animal activity in their home-cage. The potential utility of this technology remains understudied and deserves investigation in the field of oncology. Methods: Using the DVC^®^ platform, we sought to determine if the continuous assessment of locomotor activity of mice in their home cages can serve as useful digital readout in the monitoring of animals treated with the reference oncology compounds cisplatin and cyclophosphamide. SCID mice of 14 weeks of age were housed in DVC^®^ cages in groups of four and followed with standard and digital examination before and after treatment over a 17-day total period. Results: DVC^®^ detected statistically significant effects of cisplatin on the activity of mice in the short and long term, as well as trends for cyclophosphamide. The activity differences between the vehicle- and chemotherapy-treated groups were especially marked during the nighttime, a period when animals are most active and staff are generally not available for regular checks. Standard clinical parameters, such as body weight change and clinical assessment during the day, provided additional and complementary information. Conclusion: The DVC^®^ technology enabled the home cage monitoring of mice and non-invasive detection of animal activity disturbances. It can easily be integrated into a multimodal monitoring approach to better capture the different effects of oncology drugs on anti-tumor efficacy, toxicity, and safety and improve translation to clinical studies.

## 1. Introduction

The global incidence of cancer is increasing [1] and despite significant progress in treating patients with increasingly effective immunotherapies and targeted therapies, the high rate of patients with primary and secondary resistance, as well as with cancer treatment-related morbidity, remains a concern. It underscores the limited understanding of mechanisms regulating tumor resistance and an urgent unmet need for novel approaches [2,3]. Understanding the biologic processes and environmental factors underlying cancer development and its progression, as well as developing strategies to prevent and target cancer vulnerabilities with minimal toxicities are major goals of cancer research [4]. Among the preclinical models used in this research, experimental animal models make it possible to establish and confirm biological processes and the involvement of certain genes in cancer development. They are also essential tools for testing the toxicity and efficacy of new therapeutic agents and combinations [5,6]. Despite significant investments in drug development, the overall success rate of drugs in the clinic remains low, in part due to limited reproducibility and the inability to predict drug toxicity and efficacy from preclinical data [7,8,9]. In order to improve the translation of early preclinical discoveries into successful clinical studies, there is a need for continued improvement of in vitro, in situ, and in vivo preclinical models and their monitoring, while recognizing that animal studies remain a standard for establishing evidence of the link between a pharmacological target and human disease [7,8,9,10]. On the other hand, the 3Rs (replacement, reduction, and refinement) recommendations require preserving animal welfare by minimizing all forms of suffering, pain, and distress to animals during experiments. It is mandatory to improve social interactions in cages by housing animals in groups rather than individually [11]. Therefore, any solution or study design allowing group housing should be encouraged and promoted over individual housing. Animal monitoring should be adapted so that neither death nor significant pain remain experimental endpoints in preclinical studies [12,13,14]. Accordingly, abnormal behaviors, significant weight loss or a tumor reaching a certain size are considered as limiting endpoints leading to termination of the experiment. In addition, the refinement of methods to maximize the information gathered per animal in an experiment are recommended to reduce the number of animals, while ensuring the robustness and reproducibility of results. Digital tools that allow longitudinal measurements can meet these requirements.

Over the last several years, high-tech and digital devices have emerged as attractive tools for monitoring human subjects and patients. Often connected to a smartphone or reader, they have been proven useful for monitoring various physiological parameters and vital signs such as body temperature, heart rate, number of steps, respiratory rate, or blood glucose. Systems are now being developed for animal studies that could significantly improve the monitoring and detection of a particular phenotype, reduce disturbance and stress-related morbidities and the likelihood of errors, and predict certain toxicities and discomfort related to disease or treatment [15,16,17,18,19,20,21].

There are a variety of technologies with similarities and differences. Among the most commonly used systems, the DVC^®^ (digital ventilated cage) relies on technology employing an electromagnetic field to monitor the locomotion of group-housed animals [22,23], while the Vium^®^ system relies on the video recording of a single mouse in the cage measuring its locomotor activity, as well as additional parameters such as respiratory rate and voluntary running on a wheel installed in the cage [24,25,26]. Although these reports have highlighted the potential of these methods in different disease models, the benefits for preclinical oncology research have been poorly explored, and the validation of these digital systems in this context warrants further investigation.

In this study, using the DVC^®^ platform housing SCID mice treated with cisplatin or cyclophosphamide, we aimed to determine if the continuous assessment of locomotor activity of animals in their home cages can serve as a useful indicator to detect adverse effects in mice treated with oncology compounds.

## 2. Materials and Methods

### 2.1. Animals, Housing, and DVC-Based Activity Monitoring

Mice were obtained from Charles River Laboratories (Saint-Germain-Nuelles, France). The study involved a total of forty-eight congenic C17 female SCID mice (CB17/lCR-Prkdcscid/lcrIcoCrl) and is reported in compliance with the ARRIVE guidelines [27]. At receipt, animals were housed in the animal facility for at least one month in standard individually ventilated cages and at 14 weeks old, they were transferred to digitally ventilated cages (Tecniplast, Buguggiate, Italy) for a 7-day acclimatization period before receiving the treatment procedure. Mice were housed in groups of 4 per cage with ad libitum access to food and water for the duration of the study. Each cage is enriched with a cardboard tunnel for shelter, gnawing, and nesting materials to ensure compliance with the 3Rs. The mice were housed on a 12 h light/dark cycle with lights on from 6 a.m. to 6 p.m. Room temperature was maintained at 22 ± 2 °C with a relative humidity of 55 ± 15%. Cages were changed once a week, providing new nesting material. The DVC^®^ technology and procedure for activity monitoring has been reported previously [15,22,23]. Briefly, DVC was used to monitor mouse activity 24 h a day, 7 days a week. The technology is based on an electromagnetic field and a sensor board consisting of twelve electrodes installed under the cage, to detect differences in electrical capacitance (every 0.25 s) which can be transformed computationally into the Animal Locomotion Index, a quantitative metric developed by Tecniplast. The data collected were sent automatically to a web-based platform, then exported and processed for statistical analysis.

### 2.2. Study Design

Seven days before the first injection (noted Day −7), mice were transferred from individually ventilated cages to the DVC^®^ system (Figure 1). On Day 0, cages were randomized into 3 groups, without reorganization of the individuals to avoid additional stress, and mice from these groups received treatment with one of the compounds or a vehicle control on a biweekly basis for a total of four injections. The choice of sample size was based on previous studies and standard protocols carried out in-house to compare clinical and DVC^®^ parameters. Cisplatin (cis-diammineplatinum (II) dichloride) was obtained from VWR (Radnor, PA, USA) and prepared at 3 mg/kg. Cyclophosphamide monohydrate was obtained from Thermo Fisher Scientific (Waltham, MA, USA) and prepared at 150 mg/kg. All substances were dissolved in a sterile 0.9% NaCl solution (CDM Lavoisier, Paris, France) and injected intraperitoneally (i.p.) with a volume of 0.25 mL. Animals in the control group received an injection of the corresponding vehicle. The doses used were based on previously published work [28] and pilot studies evaluating the highest doses that could be administered without lethality or a decrease in body weight > 15%, while demonstrating anti-tumor efficacy in murine tumor models.

Digital recording started on Day −4 of the study. When indicated, values obtained from Day −4 to Day −1 preceding treatment were used as a baseline measurement to assess activity changes over the course of the study. The digital biomarkers were collected continuously throughout the experiment until Day 13.

### 2.3. Clinical Examination

Clinical observations were conducted every morning by trained personnel, including at weekends. Beginning on Day −4, the body weight of each mouse was measured daily at similar times of the light cycle. As for DVC^®^ monitoring, values obtained for Day −4 to Day −1, preceding treatment, were used as a baseline to calculate body weight variations over the course of the study. Food and water consumption were also checked regularly. A clinical score reflecting the welfare of the animals was calculated from Day −4 to Day 13 based on visual inspection of the animals by two experienced staff members blinded to the treatment condition. Scores estimated coat appearance and motility each morning, at the time of daily checks, from normal, active (1); shaggy fur OR altered motility (2); very shaggy fur AND hypomotility OR other severe clinical signs (3).

### 2.4. Statistical Analysis

For body weight and locomotor parameters, the baseline corresponds to the arithmetic mean of the values measured from Days −4 to −1 (baseline activity) and statistical analysis was performed post-treatment administration on values normalized to the baseline. For body weight and locomotor activity, a two-way analysis of variance (ANOVA) was performed on the change from the baseline with the factors group (fixed) and day (repeated) and their interaction from Day 1 to Day 13. It was followed by a Dunnett’s test each day to compare both treated groups to the vehicle and a Hochberg correction for each comparison to take into account the multiplicity across time. When mentioned, body weight and locomotor activity are presented as the mean percentage change in the calculated baseline ± standard deviation (SD). 

For the scoring, an ANOVA-TYPE analysis was performed on the score with the factors group (fixed) and day (repeated) and their interaction from Day 1 to Day 13. It was followed by a Bonferroni–Holm adjustment each day to compare both treated groups to the vehicle and a Hochberg correction for each comparison to take into account the multiplicity across time. Frequency was used to describe the distribution of clinical scores across each group each day. Statistical analyses were performed using SAS^®^ version 9.4 (Institute Inc., Cary, NC, USA) for Windows 10 with a 5% significance level. Figures were generated with R 4.2.0. and Prism 8.4.2 (GraphPad Software; La Jolla, CA, USA).

## 3. Results

### 3.1. Clinical Evaluation of Body Weight Change Revealed Mild Alterations after Chemotherapy Treatment

Clinical parameters such as changes in mouse body weight, as well as daily observation of the animals were monitored as the primary endpoints to assess toxicity following treatment with chemotherapeutic agents. On Day 0, cages were randomly assigned to the control and treatment groups, and followed with standard or home-cage digital monitoring until 2 weeks after treatment at night and during the day (Figure 1A). Cisplatin and cyclophosphamide are two alkyl agents which are widely used to treat cancer patients [29,30]. Like most chemotherapeutic agents with anti-neoplastic actions, there are known to have numerous adverse effects including but not limited to nausea, alopecia, or cognitive impairments [31,32,33,34,35]. Throughout the study, no mice were found dead or had to be terminated due to health conditions, confirming the relevance of the dose selection. Water consumption was also recorded via bottle weighing throughout the study, and no differences between conditions could detected. Body weight changes are presented in Figure 1B. In the control group, body weight was relatively stable with a slight and gradual increase throughout the study, whereas in the treated groups, decreases in body weight were observed without reaching the defined ethical endpoints (−15% over 3 consecutive days or −20% in 1 day). In general, cyclophosphamide appeared to have a larger effect on body weight change than cisplatin. For mice treated with cyclophosphamide, a gradual decrease in body weight was noted after the first dose until Day 4, reaching a peak reduction of 5 to 10%. This was followed by a progressive and partial recovery in body weight until the end of the experiment, at Day 13. Mice treated with cisplatin at 3 mg/kg showed a slight and gradual decrease in body weight after the first treatment until Day 4 to a maximum reduction of 5%, then a recovery of body weight was noted at the third administration, which induced another slight decrease followed by a similar recovery at the end of the experiment (Figure 1B). Statistically significant adverse effects on body weight were demonstrated for cisplatin treatment compared to the vehicle from Day 4 post-treatment to Day 13 (d4: *p* = 0.0006; d5: *p* = 0.0009; d8: *p* = 0.0012: d9: *p* < 0.0001; d10: *p* = 0.0002; d11–d13: *p* < 0.0001), except on Days 6 and 7 with results very close to significance (*p* = 0.0619 and *p* = 0.0774, respectively) (Appendix A). Treatment with cyclophosphamide demonstrated a statistically significant decrease in body weight as compared to the vehicle from Day 2 to Day 13 (*p* < 0.0001). These results show that body weight assessment remains an essential marker in oncology studies with noticeable differences to be expected depending on the agent, dosage, and duration of treatment.

### 3.2. Clinical Assessment by Staff Revealed Signs of Discomfort Following Chemotherapy Treatment, but Little Evidence of Altered Activity

Visual inspection of the animals was used to generate a clinical score estimating the coat appearance and alterations in the motility of the animals during the day. Prior to Day 10 of treatment, daily ratings of scores ranged from the normal score of one to a mild score of two, with individual scores very occasionally reaching the more severe score of three (Figure 2A,B). Scores of two were due to signs of discomfort as assessed by fur appearance, with shaggy fur noted, but no signs of reduced motility were observed. After the third administration of cisplatin, on Day 10, several scores of three were recorded, indicating both the discomfort and decreased activity of the mice (Figure 2B), the latter being slightly improved on Day 13.

For cyclophosphamide, individual scores remained below 3 in most cases throughout the study. Further analysis showed that treatment of the mice with cyclophosphamide had statistically significant effects compared to the vehicle only on Day 1, Day 5 to Day 8, and Day 11 to Day 13 (d1: *p* = 0.0066; d5: *p* = 0.0239; d6: *p* = 0.0239; d7–d8: *p* < 0.0001; d11–d13: *p* < 0.0001, and Appendix A), whereas treatment with cisplatin had statistically significant negative effects versus the vehicle on Day 1, and from Day 7 to Day 13 (d1: *p* = 0.0119; d7–d9: *p* < 0.0001; d10: *p* = 0.0021; d11–d13: *p* < 0.0001, and Appendix A). These findings suggest that this clinical scoring can be used for animal welfare assessment after chemotherapy treatment but is of limited value in assessing overall animal activity.

### 3.3. Digital Biomarker Detected Alterations of Animal Activity during Nighttime in Animals Treated with Chemotherapy

The animal locomotion index generated by the DVC^®^ platform was used to assess overall changes in animal activity in response to chemotherapy treatment. DVC^®^ data from the cages were collected as hourly measurements. Figure 3 shows the profiles of the unnormalized locomotion index values for three representative cages. In the vehicle control cages, and also in the groups of mice treated with chemotherapy agents, a peak of activity was demonstrated during the night phase (Figure 3A–C). This phenomenon is expected since nighttime is the most active period for these rodents. After treatment (arrows), variations in animal activity could be detected. This was especially the case during the night periods. The effects seemed more apparent in the cages of mice treated with cisplatin (Figure 3B) than in those treated with cyclophosphamide (Figure 3C), with variability throughout the study days. By contrast, decreased nocturnal activity as reflected by a decrease in nighttime peak activity was not observed in cages treated with the vehicle, except on Day 0 as all the groups showed reduced activity (Figure 3A).

To further study the effects of treatment and variability between cages, we analyzed nighttime and daytime data separately and generated heatmaps from the cumulative values of 12 h intervals (Figure 4A,B). In addition, 24 h or 1 h intervals are also presented (Figure 4C and Appendix A). In many cases, the activity decline was most noticeable during the night corresponding to the treatment days (Days 0, 3, 7, and 10) and following Day 7 (Figure 4A), while variations in activity were hardly visible when examining the light phase activity (Figure 4B). Heatmaps of the 24 h cumulative values were also informative, confirming the trend observed with the 12 h night intervals (Figure 4C). The cage change did not seem to have a major impact on night activity. On the other hand, we noted variations in activity between cages within the same group, and overall, the activity values of the cyclophosphamide group were lower than in the other groups.

By examining the locomotor activity patterns from data collected with 1 h intervals, we further observed that animals in the control group exhibited reduced activity in the hours following the first injection, suggestive of stress, compared with subsequent injections (Appendix A). These effects were less perceptible in cages with mice treated with the chemotherapeutic agents.

Data were then transformed into the locomotor activity change relative to baseline to further investigate the effects of the treatments and time on the activity patterns, as well as to explore potential differences between cisplatin and cyclophosphamide (Figure 5). As mentioned earlier, Days −4 to −1 were used to establish baseline locomotor activity. The curves showed important activity variations over time, especially during the night phase in the group treated with cisplatin, with changes under treatment that could go well beyond a 25% reduction in the nighttime activity, following treatment administration (Days 0, 3, 7, and 10) (Figure 5A). A gradual recovery of activity was observed after the first and second dose, but subsequent doses were accompanied by an overall impairment of activity over the following days. The largest differences were observed after Day 9, with a more than 50% decrease in activity in cages treated with cisplatin, suggesting chronic toxicity affecting this metric over the long term. This trend was also found in the 12 h light phase activity (Figure 5B) but with greater variability between cages. The curves generated from the cumulative activity over the 24 h period, in the meantime, seemed quite comparable to the curves of the 12 h night period (Figure 5C).

A post-hoc Dunnett’s test followed by a Hochberg correction for multiple comparisons further revealed statistically significant differences in the cisplatin group compared with the vehicle-treated counterparts at nighttime on Day 3 (*p* = 0.0030) and from Day 7 to Day 13 (d7: *p* = 0.0081; d8: *p* = 0.0032; d10–d11: *p* < 0.0001; d12: *p* = 0.0030; d13: *p* = 0.0038), except on Day 9 with a result very close to significance (*p* = 0.0534) (Appendix A). In contrast, the cyclophosphamide-treated cages demonstrated statistically significant differences only at Day 10 (*p* = 0.0131). This apparent larger effect of cisplatin on activity contrasts with what was found for body weight changes in the animals, where cyclophosphamide showed a larger effect. The measurements obtained during the light phase did not reveal a statistically significant effect of the treatment compared to the vehicle in either of the treated groups (Appendix A). Consistent with the clinical scoring results, this suggests that during the day, the treatment effect on animal activity is only weakly detectable using DVC^®^ technology, despite possible long-term effects being detectable over time upon repeated measurement. The combination of diurnal and nocturnal values (24 h intervals) confirms some of the results obtained for the 12 h nighttime intervals, with statistically significant differences observed from Day 7 to Day 13 in cages receiving cisplatin (d7: *p* = 0.0012; d8: *p* = 0.0034; d9: *p* = 0.0041; d10–11: *p* < 0.0001; d12: *p* = 0.0009; d13: *p* = 0.0041, Appendix A). In this setting, no statistically significant differences could be detected however for the group receiving cyclophosphamide compared to the control group.

Together, the results demonstrated that continuous DVC^®^ monitoring can accurately capture both acute and chronic activity disturbances across different chemotherapy treatments. During the night, animal activity was significantly impaired by repeated cisplatin treatment, whereas cyclophosphamide had a more limited effect on animal activity at the doses tested. The results also suggest that the 12 h assessment of nocturnal activity outperforms 24 h interval assessment.

## 4. Discussion

In this study, we investigated whether the DVC^®^ technology could be valuable during preclinical studies, particularly in the field of oncology, allowing a less intrusive and more objective assessment of physiological and behavioral characteristics in group-housed animals. The study protocol was aligned with a discovery protocol used by our research teams, including the selection of reference compounds tested, study duration, and sex of the animals (females only). In this validation study, the compounds were selected on the basis of previous validation protocols, enabling the use of a single dose (effects already identified), limiting study repetition and reducing the number of animals used. These two compounds are considered to be positive controls in the field of oncology, with strong potential for convincing research teams. DVC^®^-based monitoring was applied to mice with the common SCID background receiving repeated doses of two widely used chemotherapeutic agents: cisplatin and cyclophosphamide. Body weight measurements and clinical signs assessments, two essential tools in animal experimentation, were also performed on each mouse in parallel to assess animal welfare and toxicity. As expected, body weight measurements provided significant information throughout the study, with clear variations detected during this repeated treatment procedure, and between study groups. In this setting, cyclophosphamide administration appeared to have a greater effect than cisplatin on body weight change.

The clinical score was used to assess the discomfort of the animals. The scoring system used was deliberately simple to ensure limited heterogeneity between technicians and to match routine daily observations. Furthermore, the aim of the study was not to correlate DVC^®^ results with a reference clinical scoring system. This simple scoring, performed by trained personnel, allowed for the detection of acute effects (as early as Day 1) of the drugs, as well as chronic effects in the following days, often significant, with mild alterations (scores of two) until Day 10. A more severe clinical state was then observed on the last days of the study with an increase in scores of three, probably signifying cumulative effects. The effects of cisplatin appeared to increase progressively over time, whereas for cyclophosphamide, no significant difference was demonstrated on certain days compared to the control group receiving the vehicle (for example, Days 9 and 10). This underlines in part the limitations and subjective nature of such scoring, estimated at a specific and short time point during daily checks of animals, and in the absence of quantitative measures. Consistently, the application of the clinical score to assess signs of alterations in the global activity of the animals did not seem to be the most suitable method, since the variable that contributed most to the score was the alteration of the fur rather than a motility deficit. This suggests the lack of sensitivity of this scoring to detect disturbances in global activity. In contrast, continuous monitoring of the DVC^®^ activity metric was able to detect significant disturbances in the nighttime activity of the animals treated with cisplatin that could not be well estimated by clinical sign assessment by staff during the daytime.

The current study included twelve cages of four mice, and despite the relatively small number of cages in each group (i.e., *n* = 4 per group), which represents the value used in the statistical analysis of DVC^®^ data, continuous measurements detected significant decreases in the activity of the mice treated with cisplatin. The activity disturbances measured by DVC^®^ were consistently detectable in this group from Day 7 to Day 13, which was not the case for the cyclophosphamide-treated group, in which no significant alterations were identified, with the exception of Day 10 (Appendix A). These data suggest that cisplatin has a more chronic effect on animal activity than cyclophosphamide, which shows a more transient effect at the doses tested. In the latter group, non-significant trends of decreased activity were nevertheless observed during the repeated treatment procedure.

Of the time intervals tested in this study (24 h, 12 h light phase, 12 h dark phase), the 12 h measurement during the dark phase appeared to be the most relevant for discriminating treatment effects on animal activity and differentiating treatment groups. The cumulative 24 h activity was an interesting parameter to confirm the effects observed during the night in this mouse model, but it may lack sensitivity compared to nighttime activity alone, potentially limiting the detection of smaller effects. These observations are consistent with the previous report of Bains et al. [36]. For short-term observations, an hourly analysis is preferable, since the 12 h and 24 h intervals did not show significant differences.

One potential limitation of our study is that for comparison purposes, groups of mice were evaluated by both DVC^®^ and standard clinical assessment. It is known that repeated monitoring of mice during the day, or isolation of an animal for an experiment, can be stressful for the animals, potentially influencing the activity results collected by DVC^®^ during the light phase. Nevertheless, we envision that this multimodal strategy is best suited for future preclinical application involving drug testing.

Beyond the statistical significance of the measurements, the biological relevance of the changes detected must also be emphasized. In the cisplatin group, compared with the control group, the differences in body weight were in the range of 5–7% difference, which remains a small variation generally considered to be associated with an absence of or limited toxic adverse effects, and these were less marked than those observed with cyclophosphamide. Conversely, the activity of the animals was clearly affected by cisplatin treatment but not by cyclophosphamide. These observations argue in favor of the combined use of these methods.

It should also be noted that during the course of the study, we did not experience any serious events such as the mortality of individual animal subjects or early withdrawal due to significant toxicity, which might have resulted in a decrease in overall activity in certain cages, since DVC^®^ is able to capture the average activity in the cage but not the activity of the individual animal subjects. We would expect a relatively linear activity as long as the cages contain two to four animals. This point will have to be addressed in follow-up studies, especially in animal models developing tumors whose overall health can be rapidly reduced, with increased heterogeneity between individuals in terms of responses. Furthermore, effects on body weight are often more complex to assess in studies with tumors, and have an impact on variability between groups, with control animals gaining more tumor weight than treated animals, in whom the weight reduction may be partly due to tumor regression.

Our study using DVC^®^ technology also provided novel scientific insights. Cisplatin and cyclophosphamide are known to cross the blood–brain barrier with side effects on patients including neuropathies [35,37,38,39]. For cisplatin, studies in rats and mice, and using more traditional and specific methods, have already shown oxidative stress, impaired neurogenesis, neuronal damage, and neuroinflammation, which can lead to biochemical and behavioral disturbances particularly when repeated doses are administered [40,41,42,43,44,45,46]. These include attention deficit and decreased episodic and spatial memory [40,41,42,43,44,45,46]. It would be interesting to know whether the decrease in animal activity observed in our study can be explained by such cognitive deficits. Other behavioral or biochemical disturbances could also be at play. Moreover, we observed similar trends with cyclophosphamide treatment, although the effects appeared to be more transient at the doses tested, and overall, of a lesser magnitude. Experimental studies in mice and rats have previously reported neuroinflammation and neuronal degeneration under acute cyclophosphamide treatment [47,48], as well as learning and spatial memory deficits under repeated treatment conditions [49,50,51].

The SCID mice used in our study have a severe combined immunodeficiency affecting both B and T cells. They are generally used as hosts for the xenotransplantation of human cancer cell lines, patient-derived tumor xenografts or mouse tumor cells and are commonly used for safety and efficacy studies in oncology but less often for behavioral studies. Further studies using different models and other compounds are needed to further investigate the findings.

## 5. Conclusions

DVC^®^ technology demonstrated utility in this study to enhance current approaches and complement daytime observations by staff in order to capture the deleterious effects of compounds on activity behavior, toxicity, and degradation of animal welfare over the long term. Such disturbances in animals, if confirmed, should be considered sufficient signs of morbidity to establish a cut-off point and endpoints for discontinuing experiments when necessary. Additional tools under development, compatible with DVC^®^ or other systems using artificial intelligence, should help to confirm the findings and better define the behavioral perturbations under acute and intermittent treatment procedures, while distinguishing the effects of single agents and combinatorial approaches [26,36,52]. We propose that these digital tools be given greater consideration in particular in preclinical oncology studies in order to avoid misinterpretation in preclinical drug development and research studies, providing useful information through continuous monitoring that is often superior to intermittent assessment and requires minimal staff intervention. Importantly, these technologies can provide a more holistic picture of drug effects and disease phenotyping than effects solely on tumor growth and body weight, incorporating wellness, pain, and quality of life as fundamental aspects of preclinical assessment to improve translation to the clinic. Furthermore, regardless of the research field and study type, improving clinical signs’ and associated ethical endpoints’ detection is always beneficial to animal welfare.

## Figures and Tables

**Figure 1 cancers-15-04798-f001:**
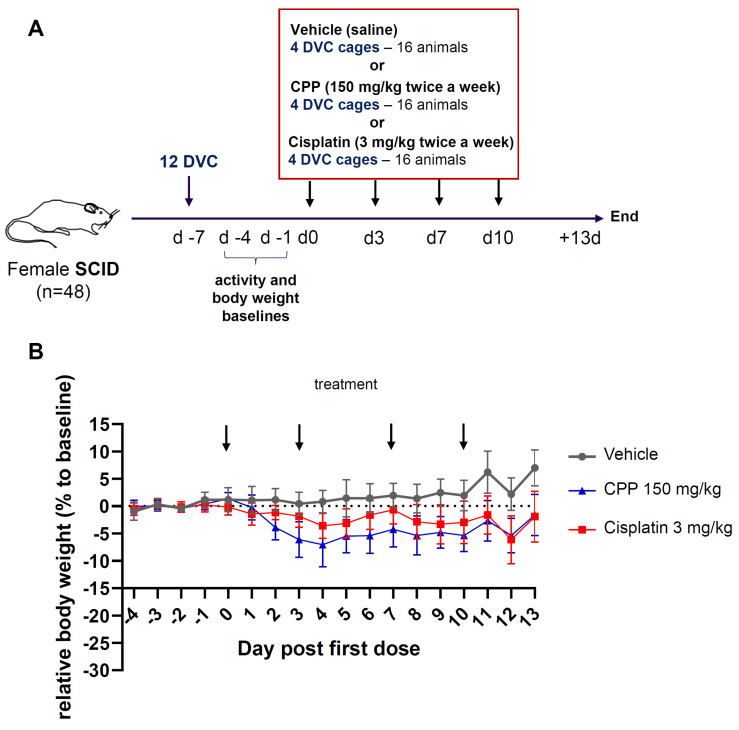
Design and body weight measurements. (**A**) A schematic representation of the study design. After 7 days of acclimation, mice in DVC^®^ systems were randomized into three study groups receiving the indicated agent or vehicle. Digital biomarkers (DVC^®^ locomotion index) and standard clinical information (body weight, appearance) were collected during the study. (**B**) Changes in body weight. Body weight was measured from Day −4 and every day for all three groups of mice. Relative weight changes from the baseline ± SD are shown. (two-way RM ANOVA; group *p* < 0.0001, day *p* < 0.0001, group × day *p* < 0.0001). The arrows indicate days of treatment.

**Figure 2 cancers-15-04798-f002:**
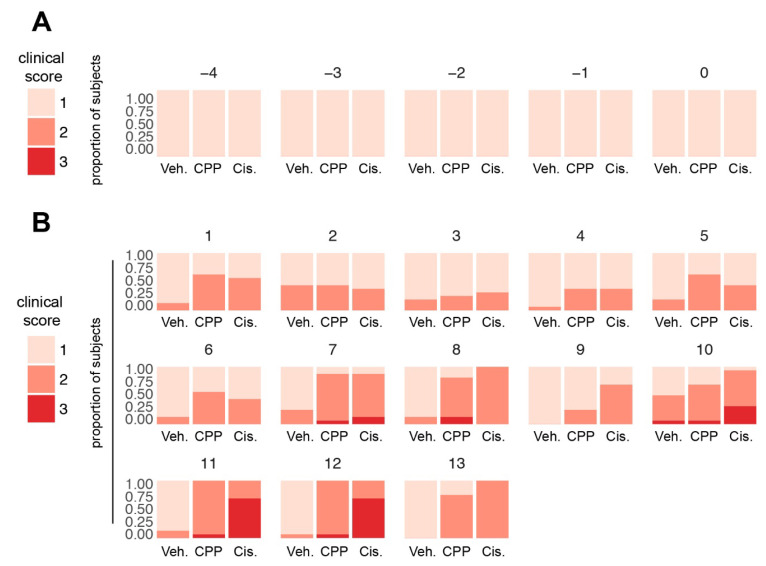
Distribution of the clinical scores in the three study groups (*n* = 48) for (**A**) days corresponding to the baseline activity (Days −4 to −1) plus the day of the first treatment administration, Day 0; and (**B**) days following the first treatment administration. A two-way ANOVA-TYPE was applied to assess global effects with treatment group, day and group x day interaction; *p* < 0.0001, followed by a Bonferroni–Holm adjustment for each day and a Hochberg correction for each comparison to account for multiplicity across treatment and time. These statistics are summarized in Appendix A. Veh., vehicle; CPP, cyclophosphamide; Cis., cisplatin.

**Figure 3 cancers-15-04798-f003:**
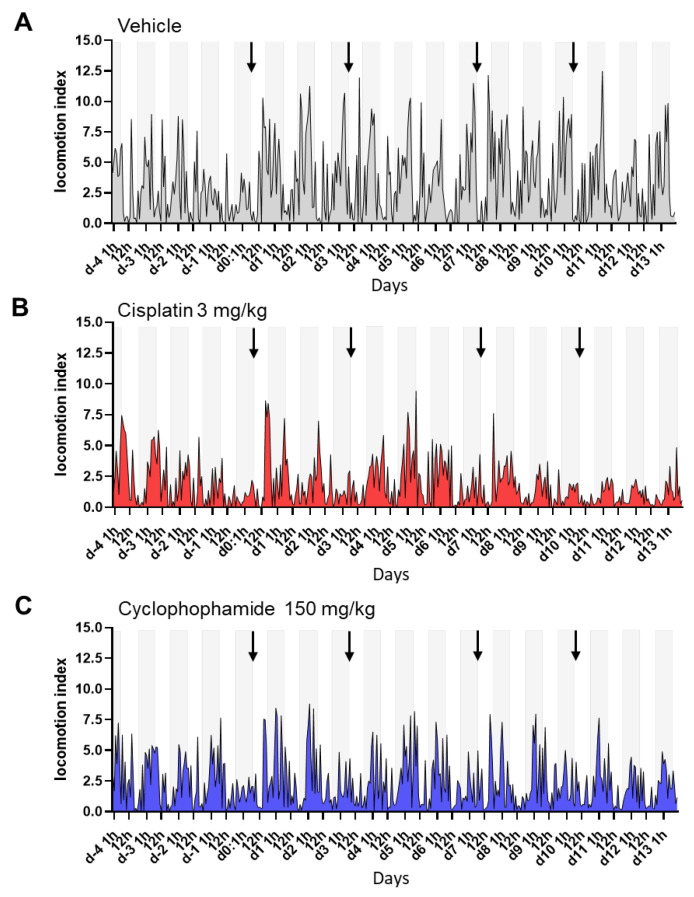
DVC^®^ locomotion index profile revealed alterations in animal activity upon chemotherapy intervention. Locomotion activity profiles generated from locomotion indexes in representative group-housed DVC^®^ cages from each treatment group: (**A**) vehicle, (**B**) cisplatin, (**C**) cyclophosphamide. Values were calculated from measurements collected over 1 h intervals. The gray areas correspond to the dark phase. The arrows indicate the days of injection.

**Figure 4 cancers-15-04798-f004:**
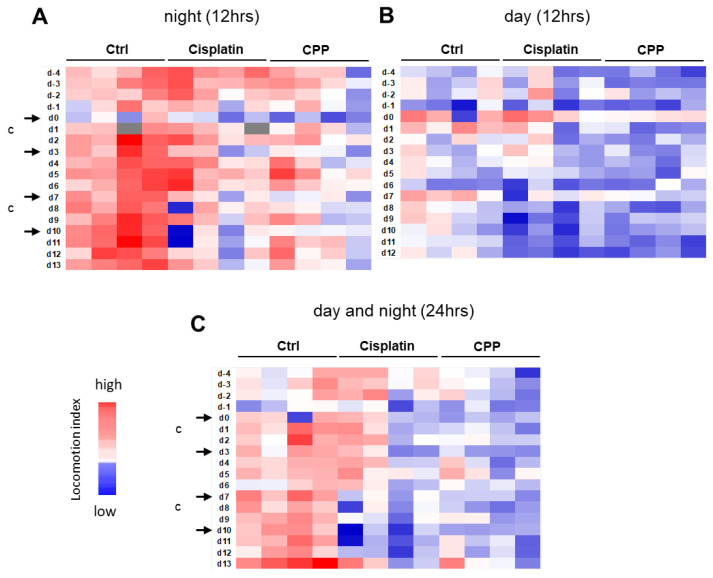
DVC^®^ locomotion patterns confirmed alterations of animal activity upon chemotherapy intervention. (**A**) Heatmap of the dark phase from Days −4 to 13 for the three study groups. The heatmap shows activity data in the indicated groups with the locomotion index displayed in 12 h intervals (each row) during lights-off. Each column represents a DVC^®^ unit housing 4 mice. Colors denote the levels of calculated locomotion index, from blue (reduced locomotion index) to red (high locomotion index) (**B**) Heatmap of the light phase on Days −4 to 13 for the three study groups in 12 h intervals (**C**) heatmap depicting the full day activity (24 h intervals). The gray color indicates time slots for which no data were collected. Arrows indicate treatment intervention. Days of cage change “c” are specified.

**Figure 5 cancers-15-04798-f005:**
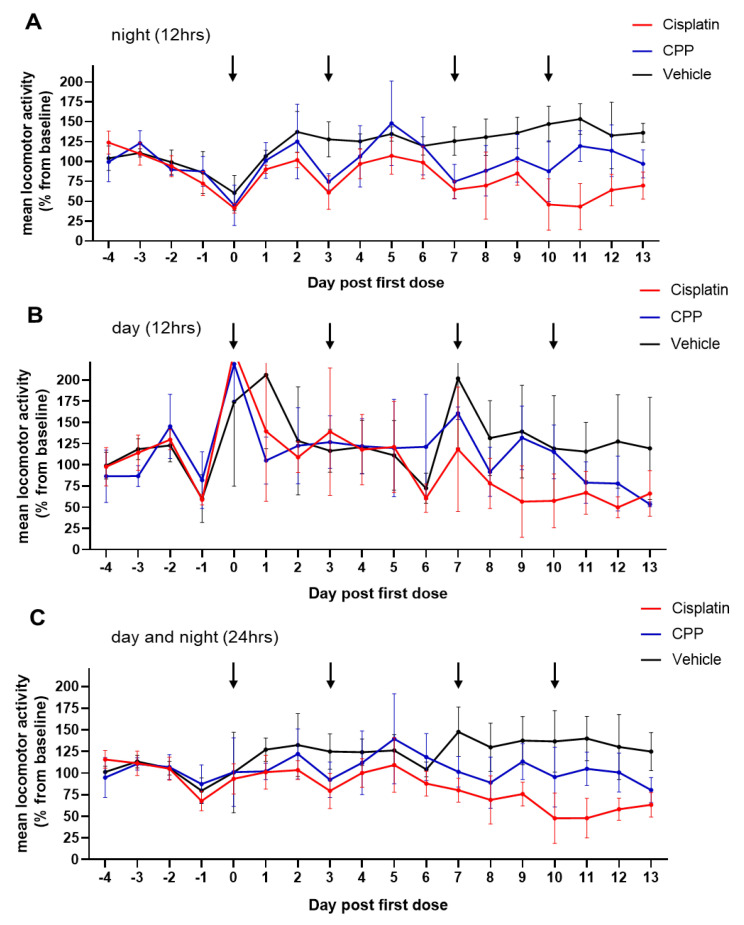
Locomotion activity change revealed significant adverse effects of cisplatin on animal activity. Curves depicting the nocturnal and diurnal activity (night and day time activity). Locomotion activity change is presented as a percent of baseline ± SD, (**A**) at night (two-way ANOVA; study group (*p* < 0.0001), time (*p* = 0.0002), time by study group interaction (*p* = 0.0438)); (**B**) daytime (two-way ANOVA; study group (*p* = 0.1204); time (*p* < 0.0001), time by study group (*p* = 0.0317)); or (**C**) the entire day (two-way ANOVA; study group (*p* < 0.0001), time (*p* = 0.0002), time by study group interaction (*p* = 0.05)). Statistical comparisons for the different days are summarized in Appendix A.

## Data Availability

The data presented in this study are available on request from the corresponding author.

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
