# Peer review of "Activity in Group-Housed Home Cages of Mice as a Novel Preclinical Biomarker in Oncology Studies"

_cancers, 2023, doi:10.3390/cancers15194798_

Round 1

Reviewer 1 Report

    1. In this research, the authors examined the potential of digital ventilated cage technology for tracking locomotor activity. The primary objective of the study was to determine whether the activity within home cages where animals were housed together could serve as a preclinical biomarker.
    2. My comments are as follows:
    3. 1. The authors must provide a definition of SCID in the abstract. 2. The authors should explain the reasoning behind exclusively including females in the study. 3. Although the Cisplatin group experienced a notable decrease in body weight, there appears to be no substantial alteration in health based on the clinical scoring. It would be beneficial for the reader if the authors provided more details regarding the clinical scoring methodology. Furthermore, future studies could benefit from the integration of a more comprehensive health assessment method, such as a 12-point scoring system, which evaluates various facets of animal health. 4. What was the rationale for incorporating only a single dose of Cisplatin and CPP in the study? 5. In order to more accurately assess the importance of this model, it is advisable to incorporate a positive control group, or alternatively, provide a discussion on why a positive control might not be essential for drawing conclusions.

  1.  

Author Response

Comment 1: The authors must provide a definition of SCID in the abstract.

Answer: We thank the reviewer for bringing this to our attention.  We have addressed this point in the revised abstract (line 20)

Excerpt from the manuscript: …(Severe Combined Immunodeficiency)…

Comment 2. The authors should explain the reasoning behind exclusively including females in the study.

Answer: We have addressed this point in the first paragraph of the discussion (lines 355-357)

Excerpt from the manuscript: “The study protocol was aligned with a discovery protocol used by our research teams, including selection of reference compounds tested, study duration and sex of animals (females only)”

Comment 3. Although the Cisplatin group experienced a notable decrease in body weight, there appears to be no substantial alteration in health based on the clinical scoring. It would be beneficial for the reader if the authors provided more details regarding the clinical scoring methodology. Furthermore, future studies could benefit from the integration of a more comprehensive health assessment method, such as a 12-point scoring system, which evaluates various facets of animal health.

Answer: We thank the reviewer for this constructive remark. We have touched on this point in the third paragraph of the discussion (lines 370-373)

Excerpt from the manuscript: “The scoring system used was deliberately simple to ensure limited heterogeneity between technicians and to match routine daily observations. Furthermore, the aim of the study was not to correlate DVC® results with a reference clinical scoring system”

Comment 4. What was the rationale for incorporating only a single dose of Cisplatin and CPP in the study?

Answer: We thank the reviewer for this comment and the opportunity to clarify. This request has been addressed in the first paragraph of the discussion (lines 357-361)

Excerpt from the manuscript: ”The study protocol was aligned with a discovery protocol used by our research teams, including selection of reference compounds tested, study duration and sex of animals (females only). In this validation study, the compounds were selected on the basis of previous validation protocols, enabling the use of a single dose (effects already identified), limiting study repetition and reducing the number of animals used. These two compounds are considered to be positive controls in the field of oncology, with strong potential for convincing research teams…”

Comment 5. In order to more accurately assess the importance of this model, it is advisable to incorporate a positive control group, or alternatively, provide a discussion on why a positive control might not be essential for drawing conclusions.

Answer: We thank this reviewer for his/her valuable comment. This concern was also addressed in the above paragraph of the discussion in the revised manuscript (lines 359-361)

Excerpt from the manuscript: ” …These two compounds are considered to be positive controls in the field of oncology, with strong potential for convincing research teams”

Reviewer 2 Report

This study is a well-written manuscript presenting a novel pre-clinical parameter for assessing the clinical status of mice during chemotherapy treatment. The study demonstrates that subtle changes in activity can be detected during night with the help of digital activity monitoring sing DVC, as a result of treatment with chemotherapeutical agents, even though no changes are observed in body weight.

Suggestions for minor changes are indicated in the MS. These include mainly few grammatical suggestions and some sections in the results which should be moved to the discussion. The last line of the conclusion could include an additional benefit of DVC contributing to increased animal welfare in pre-clinical studies through early detection of endpoints.

Author Response

This study is a well-written manuscript presenting a novel pre-clinical parameter for assessing the clinical status of mice during chemotherapy treatment. The study demonstrates that subtle changes in activity can be detected during night with the help of digital activity monitoring sing DVC, as a result of treatment with chemotherapeutical agents, even though no changes are observed in body weight.

Answer: We thank the reviewer for this positive feedback

Comment 1: These include mainly few grammatical suggestions and some sections in the results which should be moved to the discussion.

We appreciate the reviewer’s suggestion. We have corrected a few typos. For sake of clarity and because we have already greatly expended the discussion in response to the reviewers’1 suggestions. We have respectfully decided to keep the results sections as they were in the original version

Comment 2: The last line of the conclusion could include an additional benefit of DVC contributing to increased animal welfare in pre-clinical studies through early detection of endpoints.

Answer: We thank the reviewer for his/her suggestions. Accordingly, we have extended the conclusion part with these comments (lines 472-473)

Excerpt from the manuscript: ”Not to mention that, regardless of the research field and study type, improving clinical signs' and associated ethical endpoints’ detection would always be beneficial to animal welfare”

Round 2

Reviewer 1 Report

The comments have been addressed and the manuscript can be accepted for publication.